# Effect of Different Operational Conditions on the Treatment Performance of Milk Processing Wastewater (MPW) Using a Single Stage Flexible Fibre Biofilm Reactor (SS-FFBR)

**DOI:** 10.3390/membranes13010037

**Published:** 2022-12-28

**Authors:** Mohamed Abdulgader, Jimmy Yu, Ali Akbar Zinatizadeh, Philip Williams, Zahra Rahimi

**Affiliations:** 1School of Engineering and Built Environment, Griffith University, Nathan Campus, Nathan, QLD 4111, Australia; 2Department of Environmental Science, Faculty of Environment & Natural Resources, Wadi Al-Shatti University, Brack Al-Shatti, Libya; 3Department of Applied Chemistry, Faculty of Chemistry, Razi University, Kermanshah 6718773654, Iran; 4Environmental and Pollution Engineering Group, Environmental Research Center (ERC), Razi University, Kermanshah 6718773654, Iran

**Keywords:** attached growth reactor, SS-FFBR, milk processing wastewater

## Abstract

The performance of a biofilm system, single-stage flexible fibre biofilm reactor (SS-FFBR) treating milk processing wastewater (MPW) is evaluated under various process and operational conditions. The system behavior is analyzed with different biological and physical parameters. Results show that the high COD removal efficiency of 95% is obtained at a low COD_in_ concentration of 809 mg/L. However, the COD removal is slightly decreased to 91.7% once the COD_in_ concentration incremented to nearly 4000 mg/L. The effect of organic loading rate (OLR) on the SS-FFBR performance is examined as total suspended solids removal efficiency, dissolved oxygen, and turbidity. The SS-FFBR showed considerable performance, so that 89.9% and 89.7% removal efficiencies in terms of COD and TSS removals, respectively, obtained at the highest OLR of 11.7 kg COD/m^3^d. TSS removal efficiency of 96.7% is obtained at a low OLR of 1.145 kg COD/m^3^d. A linear relationship between the OLR and COD removal rate was revealed. The COD removal rate was incremented from 1.08 to 10.68 kg COD/m^3^d as the OLR increased from 1.145 to 11.7 kg COD/m^3^d. Finally, the operating system is a promising technique recommended to treat various industrial wastewaters with high OLR.

## 1. Introduction

Nowadays, the imbalance between availability and consumption of drinking water has led to the incidence of extreme water scarcity in some parts of the globe. On the other hand, the water quality is worsening owing to increasing environmental and industrial activities. Therefore, the supply of sustainable and safe water sources, as well as aquatic environment protection, have become one of the substantial issues in today’s society. Recently, reclamation of wastewater with the aim of water reuse has attracted much attention from researchers. The usage of biofilm, defined as communities of microorganisms attached to support media, has become a hot research issue in the perspective of wastewater remediation through biofilm reactors. The biofilms as principal biocatalysts transform biodegradable organic compounds and nutrients regarded as environmental contaminants into harmless products [1].

Compared to suspended growth reactors, the employment of biofilm reactors in wastewater treatment renders better biological treatment performance via high biomass concentrations and much more resistance to organic shock loads [2]. Moreover, there are no challenges related to sludge handling and bulking phenomenon common in the case of many suspended growth systems [2]. Overall, numerous configurations of the biofilm reactors have been used to treat municipal and industrial wastewaters. However, the majority of studies are focused on moving bed bioreactors (MBBRs) [3], integrated fixed-film activated sludge (IFAS) processes [4], membrane biofilm reactors (MBfRs) [5], and rotating biological contactors (RBC) [6]. The biofilm reactors, despite their abundant usefulness, may be undesirable in biological treatment operation and arise operational difficulties that increase the treatment costs [1,7]. The use of flexible fibre media in the biofilm reactors, called flexible fibre biofilm reactor (FFBR), can reduce the operational expenses via the low cost of packing media, elimination of recycling sludge stream as a result of low washout rate, and provision of very high specific surface area (~2200 m^2^/m^3^) [7]. Additionally, installation of flexible fibre bundles inside the reactor is easy and also pressure loss is negligible in these reactors [7]. Compared to other biofilm reactors, blocking problems are avoided in the FFBR due to the flexibility of packing media and movements induced by air and water flows [8]. Therefore, these prominent advantages have encouraged the researchers to develop the FFBR in different configurations to treat high strength wastewaters. In this sense, Yu et al. developed FFBR into a two-stage flexible fibre biofilm reactor to treat food processing wastewater [9]. Total chemical oxygen demand (COD) removal efficiency was 97% at COD_in_ of 2700 mg/L with an organic loading rate (OLR) of 7.7 kg/m^3^d. With the inspiration from this work, we designed an innovative sequencing batch flexible fibre biofilm reactor (SB-FFBR), flexible fibre biofilm bioreactor, and multistage flexible fibre biofilm reactor to remediate raw milk processing wastewater [10,11,12,13]. The COD and TSS removal efficiencies were reported to be 92.5% and 89.1%, respectively, showing the high performance of the biofilm reactor. In another study, we obtained high COD removal efficiency of 97% at 1.6 d hydraulic retention time (HRT) and 2.74 kgCOD/m^3^d OLR using the same reactor and wastewater at different operational conditions [14]. Zinatizadeh et al. investigated the performance of a moving bed biofilm reactor (MBBR) treating municipal wastewater under various operating conditions as HRT (4–12 h) and DO concentration (2–4 mg/L) [3]. The maximum COD removal efficiency of 88% was obtained at HRT of 12 h and AFR of 4 mg/L using Kaldnes-3 as packing media. Mirghorayshi et al. studied the bioprocess functionality of a novel hybrid airlift bioreactor (HALBR) treating composting leachate at various HRTs, AFRs, and AVRs of 18–30 h, 1–2 L/min and 0.22–0.26, respectively [15]. The COD removal efficiency of 90% was reported at optimum region i.e., HRT of 28.3–30 h and AFR of 1.7–2 L/min.

In our previous work, a single-stage flexible fibre biofilm reactor (SS-FFBR) was employed to treat milk processing industrial wastewater [8]. Process analysis and optimization of the proposed system were studied using response surface methodology (RSM) at various HRTs (8 h, 12 h, and 16 h) and COD_in_ (836 mg/L, 763 mg/L, 2477 mg/L, 2480 mg/L, 2266 mg/L, 3922 mg/L, and 4010 mg/L). Accordingly, in the present research, we intend to study the biological performance of SS-FFBR working under different process and operational conditions for the treatment of high strength milk processing wastewater (MPW). To achieve this aim, the effect of OLR, COD_in_, solids loading rate (SLR), and hydraulic retention time (HRT) are evaluated on the effluent parameters. In this work, we envisage the proposed system would be able to produce a high effluent quality.

## 2. Materials and Methods

### 2.1. Feedstock Preparation and Inoculum Culture

Feedstock was real milk processing wastewater collected from the final collection wastewater basin located at National Foods Milk Ltd., Brisbane, Queensland, Australia. Samples were maintained in a 0–4 °C refrigerator before the use. The controlled parameters in the feedstock were the organic loading rate (OLR), chemical oxygen demand (COD), and solids loading rate (SLR). OLR, COD, and SLR were prepared in predetermined amounts by diluting raw wastewater. The characteristics of feedstock are presented in Table 1. The inoculum seed was comprised of a mixture of two activated sludge samples obtained from Oxley Creek Wastewater Treatment Plant and Loganholme Wastewater Treatment Plant (Brisbane, Australia). The SS-FFBR was inoculated by a one L-volume of the mixed culture.

### 2.2. Experimental Set-Up and Operational Strategy

A lab-scale single stage flexible fibre biofilm reactor (SS-FFBR) was fabricated and set up at the microbiology science laboratory, Griffith School of Engineering, Griffith University, Australia. The reactor set-up is depicted schematically in Figure 1. A square-shaped reactor with an effective capacity of 8 L and dimensions of 650 × 125 mm (height × width) was made of transparent sheets of acrylic plastic with 6 mm thickness. The wastewater was introduced from the top of reactor using a peristaltic pump (Cole Parmer, Vernon Hills, IL, USA, Masterflex, Model 7523-60) and the treated effluent was discharged from the reactor base. The reactor was followed by a sedimentation tank with a 12.5 L volume and various HRTs. Seven flexible fibre bundles as packing media were circularly fixed in the centre of the reactor with the aid of a rope. The flexible fibres were attached to the rope at intervals of approximately 80 mm. The specifications and arrangement of the packing media inside the reactor are presented in Table 2 and Figure 2, respectively. The highlighted feature of the flexible fibres is their large surface area which provides a large space for growth and attachment of the microorganisms. The reactor was aerated and mixed using two air diffusers fixed at the base of the reactor.

The reactor was started up to develop the biofilm growth and be acclimatized with new substrate. Initially, the reactor was inoculated with a 500 mL volume of activated sludge with a volatile suspended solid (VSS) concentration of 4189 mg/L and VSS/SS ratio of 0.79. Then, the reactor was daily fed up as batch wise with 500 mg/L COD containing wastewater. The COD concentration was incremented to 2417 mg/L gradually. During the start-up period, parameters of COD removal, pH, turbidity, VSS, and DO were monitored based on the Standard Methods [16]. After start-up and biofilm development for a 7-day period, experiments were continuously conducted with an initial condition of 2400 mg/L COD, 3.6 g/L d OLR and 16 h HRT. Feed pH was kept constant at 7 using hydrogen chloride and sodium hydroxide solutions throughout the experiments. In this step, the reactor reached steady state conditions after 20 days. In order to study the effectiveness of the single stage attached growth reactor on the treatment performance, the influence of four operating and process variables, listed in Table 3, was investigated. In each experimental run, sludge concentration in the reactor was monitored. Dissolved oxygen (DO) concentration inside the reactor was more than 2 mg/L. The performance of the SS-FFBR was evaluated in terms of COD removal efficiency/rate, TSS removal efficiency/rate, and VSS removal calculated by the Standard Methods [16]. The sampling was obtained from the supernatant of the settling tank. The dissolved oxygen (DO) concentration was monitored using a DO probe (YSI model 5010). The pH was determined by means of a pH meter (model 90-FL) supplied by TPS Pty Ltd. (Brendale, Australia). Turbidity was measured by a turbidity meter (model 2100A) provided from HACH Company (Loveland, CO, USA). All experiments were conducted at ambient temperature.

## 3. Results

### 3.1. SS-FFBR Start-Up Data

Figure 3a shows the hybrid reactor performance during the start-up period based on COD removal. The response was measured three times, and average value was reported. In the first week, the reactor was fed step-wise with an influent COD concentration varying from 500 mg/L to 800 mg/L, at an HRT of 12 h. In this period, the reactor recorded a high COD removal efficiency of approximately 80% despite the decline of the removal efficiency in some periods. Apparently, the gradual increase in the influent COD concentration resulted in a reduction of the COD removal efficiency that can be observed from day 9 to 18 (Figure 3a). It revealed that the reactor was under stress due to organic loading shocks that may affect the attached microorganisms and cause the washout of biomass. In the new environment, the adaptation of microorganisms took time before they resumed their pre-shock treatment efficiency. As can be seen in the Figure 3, from day 19 the removal efficiency increased as the COD concentration increased and the removal efficiency remained at an almost constant range around 80%, indicating a relatively stable reactor performance.

During the start-up phase, the turbidity and pH of the reactor effluent were monitored regularly. It has been established that pH is an important parameter influencing the performance of biological processes [17]. Figure 3b demonstrates the pH variations and effluent turbidity during the start-up period. The pH remained at almost neutral, in the range of 7–7.4, in the first 20 days of the start-up period. However, a significant increase in the pH value over 7.5 was observed when the influent COD concentration increased and reached a value over 2250 mg/L. However, this range of pH is still in the appropriate range for the biological process. 

The effluent turbidity showed significant variations and fluctuations. In the first week, when the influent COD concentration was below 800 mg/L, the effluent turbidity was stable at a low range between 12.5 NTU and 18 NTU. The effluent turbidity started to increase to more than 30 NTU as the influent COD concentration increased to approximately 1000 mg/L. This was due to the increase in the solids loading rate (SLR) and organic loading rate (OLR) in the system. It is also evident that because of the shorter SRT at the start-up stage, there was insufficient time for VSS hydrolysis and subsequent digestion of the organic matter. The increase in the amount of total solids in the influent wastewater may also have increased the biomass growth rate in the reactor, which contribute for the increase in the effluent turbidity by biomass washout. 

The influent and effluent VSS concentration variations during the start-up period are shown in Figure 3c. It can be seen that the effluent VSS concentration increased gradually with operational time as influent VSS concentration increased. In the first 7 days of the acclimatization, the effluent VSS concentration was low as the biofilm was in the developmental stage. The influent solid concentration applied to the reactor was also low. However, the effluent VSS concentration showed a gradual increase even when there was fluctuation on the influent VSS concentration in the second week. On day 17, the effluent VSS concentration increased significantly even when the influent VSS decreased. This could be due to the detachment of biofilm from the media, or increased growth of suspended growth microorganisms. From day 19 to day 23, the influent VSS was increased and kept constant, and in this period, the effluent VSS concentration was dropped and fluctuated, which may be attributed to increases in the sloughing rate as well as the rate of hydrolysis. 

Since the treatment process is aerobic, the DO concentration in the reactor was monitored during the start-up period. Figure 3d presents the variation in DO levels and influent COD concentrations during the start-up time. According to the figure, as the influent COD increased, the DO consumption also increased due to higher COD removal at a constant rate of aeration. During the start-up period, the DO level was continuously measured up to day 17, when the DO concentration was significantly decreased to nearly 2 mg/L. Such a decrease may be owing to the enhanced biomass growth rate in the reactor or the aeration deficiency in the system. In contrast, a high level of DO was observed on day 12 (up to 5.8 mg/L). After day 17, the DO was not measured due to a technical difficulty. However, aeration was continued as required.

Although the start-up period was run for 23 days, the time needed for the acclimatization was not as long. It was observed that the biofilm appeared quickly on the packing media and the colour changed on the surface of the media from white to brown, indicating the initial biofilm development process. Nevertheless, the start-up period of the SS-FFBR was continued for 23 days to ensure that the biofilm was completely developed and to avoid the consequences of failure of the operation. At the end of the experiments, the system reached a stable condition with an average COD removal efficiency of 77.6%, corresponding to an average OLR of 4.8 kg COD/m^3^d and HRT of 12 h. A low turbidity and VSS were also achieved. The results in this period show an improvement in the COD removal for milk processing wastewater (MPW) treatment by using a SS-FFBR reactor. 

### 3.2. Bioreactor Performance

#### COD Removal Efficiencies 

The performance of the SS-FFBR for COD removal efficiency was influenced by the influent COD concentration of the wastewater. Figure 4a presents the COD removal efficiencies at 8 h HRT for various COD concentrations. The response was measured three times, and average value was reported. It can be observed that the COD removal efficiencies decreased with increases in the influent COD concentration of the wastewater. At the stable condition, the achieved COD removal efficiencies were 95.5%, 90.3%, and 89.9% when the reactor was operated with an average influent COD concentration of 836 mg/L, 2480 mg/L, and 3922 mg/L, respectively. There was 9.1% of COD not removed and this is considered as the non-biodegradable portion of the waste organic. It is evident from the results that the SS-FFBR reactor showed consistently good performance at higher influent COD concentrations and was stabilized relatively quickly. This was mainly due to the increased biomass concentration in the reactor, which was nearly 3465 mg VSS/L in the form of suspended biomass. Akhbari et al. acquired the highest COD removal of 93.55% at HRT of 18.7 h using an integrated system (RBC-AS), rotating biological contactor (RBC), and activated sludge (AS) in treating synthetic wastewater [18].

In a study carried out by Huang and his co-workers in a membrane moving bed bioreactor (MBBR), COD removal of around 58% was obtained at HRT of 96 h [19]. In our previous study, the maximum total chemical oxygen demand (TCOD) removal of 96.6% was obtained at 12 h HRT and 2477 mg/L COD_in_ [8]. The results showed that the COD removal percentage increased by the increasing in HRT and decreasing in the COD_in_. Figure 4b illustrates the influence of COD concentration on the COD removal efficiency as a function of time at 12 HRT. At the stable condition, the reactor achieved 95% of COD removal efficiency at an average influent COD concentration of 830 mg/L, whereas 96.6% of COD removal was achieved when the influent COD concentration was increased to 2477.3 mg/L. However, the COD removal efficiency decreased to 91% when the influent COD concentration was increased to 4010 mg/L. The experimental results demonstrated that the effect of COD on removal efficiency was significantly related to the influent COD concentration. Therefore, the overall performance of the SS-FFBR was quite satisfactory even with a high influent COD concentration of wastewater. Such a high COD removal was attributed to a high concentration of biomass ranged between 1470 and 3370 mg VSS/L as suspended biomass concentration, and also almost 5000 mg VSS/L as attached biomass. Zinatizadeh and his co-workers reported maximum COD removal of 86% at 12 h HRT [3].

Figure 4c shows the COD removal efficiency at 16 h HRT as a function of time for different influent COD concentrations. At the stable condition, the SS-FFBR reactor achieved 94% of COD removal at 763 mg/L of COD influent concentration. While at 2266 mg/L of influent COD, the removal efficiency showed a slight increase to 95%. However, the removal efficiency was slightly lowered to 94.5% when the reactor operated with an influent COD concentration of 3750 mg/L. The reactor exhibited a good COD removal performance even with the increase in the influent COD. It was obvious that the reactor has a high capacity to treat high strength wastewater. The above results indicate that the SS-FFBR was able to treat raw milk processing wastewater efficiently. The main reason for such performance was due to the presence of the flexible fibre packing media, which provided a high surface area for the attached growth of microorganisms. The total biomass concentration in the reactor was estimated to be approximately 7000 mgVSS/L, with 1392 to 2060 mg VSS/L in the form of suspended biomass.

It can be inferred that the SS-FFBR is readily capable of treating high strength raw milk processing wastewater, which has not undergone any pre-treatment. However, the performance of the SS-FFBR varies with different operating conditions. The reactor’s performance is increased with an increase in the HRT from 8 to 16 h. This system indicated much better performance than the one tested by Yu et al. [7], treating food processing wastewater. The authors achieved 76% COD removal efficiency in one stage.

### 3.3. Effluent Qualities

The effects of the operating parameters on the SS-FFBR effluent quality are shown in Table 4. It is clearly shown that the qualities of the effluent decreased with an increase in the HRT or decrease in the OLR. At a low influent COD concentration of 763 to 836 mg/L, a good effluent quality was obtained with COD concentration ranging from 41 to 47.7 mg/L, and the effluent TSS concentration in the range from 5 to 15 mg/L. In addition, a very low effluent turbidity was obtained which was in the range from 2.73 to 6.46 NTU. The pH level remained between 7.5 and 7.65 throughout the experiments. At all HRTs, the system was sufficiently aerated, and the DO level was maintained above 5 mg/L, which enhanced the biomass growth. At an HRT of 8 h, the DO level reached 6 mg/L which may be because of the improved solubility of oxygen and also decreased biomass growth activity. In a study carried out by Zinatizadeh et al., maximum TSS removal of 89% was obtained at HRT of 4 h [3].

At an influent COD concentration between 2266 and 2480 mg/L, the effluent quality decreased with the decrease in HRT as shown in Table 3. The effluent COD of the SS-FFBR system increased with the decreased HRT. However, the effluent COD concentration was reported to be 83.2 mg/L at 12 h HRT while it showed a higher effluent COD concentration of 238.5 mg/L under the lowest HRT and OLR of 8 h and 7.44 kg COD/m^3^d, respectively. As the HRT increased, the effluent TSS concentration decreased except at 12 h HRT, at which the lowest TSS concentration of 20 mg/L was recorded. With respect to the effluent turbidity, the reactor showed a good effluent turbidity of 6 NTU at 12 h HRT, while it increased to 36.2 NTU when the HRT reached 8 h. The DO level in the reactor was at the recommended level and ranged from 3 mg/L to 3.45 mg/L. However, the pH value was significantly higher and reached 7.9, while it dropped to 7.73 when the HRT decreased to 8 h, which corresponds to an OLR 7.44 kg COD/m^3^d. This indicated that the reactor effluent quality was affected by HRT as well as the OLR.

When the influent COD concentration increased from 3750 to 4010 mg/L, the variation in SS-FFBR reactor effluent qualities was more obvious. The reactor achieved a good quality effluent despite the high OLR applied to the system. For example, at 16 h HRT, the effluent COD was reported to be 211.6 mg/L, while the effluent COD concentration increased to 352.8 mg/L and 394.7 mg/L at HRTs of 12 h and 8 h, respectively. The effluent turbidity increased from 28.3 to 45.5 NTU when the HRT decreased from 16 h to 8 h. The turbidity of the treated wastewater at various HRTs was correlated with the level of TSS in the effluent stream. The effluent TSS concentration also increased from 23.3 to 61.2 mg/L when the OLR increased from 5.62 to 11.7 kg COD/m^3^d. The increase in TSS concentration was probably initiated by the higher death rate of microorganisms due to the increase in the organic loading rate. The DO value in the reactor was kept at a satisfactory level and ranged between 2.28 and 3.6 mg/L. In our previous work, incremental increases in HRT and reductions in COD_in_ have a decreasing effect on the effluent turbidity [8]. The lowest effluent turbidity of 0.43 NTU was obtained at an HRT of 12 h and COD_in_ of 830 mg/L. In another study, the maximum TCOD removal efficiency was obtained at an HRT of 12 h and COD_in_ of 1590 mg/L [10]. The authors demonstrated that the increase in COD_in_ led to the reduction in contribution of consecutive stages in TCOD removal efficiency [11]. In another study, effluent turbidity of 1 NTU was reported at HRTs of 4 and 8 h [3].

### 3.4. Effect of Organic Loading Rate on Reactor Performance

The effect of OLR on the SS-FFBR performance for COD removal efficiency was evaluated by decreasing the HRT stepwise from 16 h to 8 h at different COD concentrations. Figure 5 shows the relationship between the COD removal efficiency and OLR at stable conditions. The response was measured three times, and average value was reported. The reactor showed a reverse linear relationship between the OLR and COD removal efficiency. The better performance was achieved as the lowest OLR and HRT were employed, whereas the COD removal efficiency trend was not pronounced at the HRT of 16 h and OLR of 5.62 kg COD/m^3^d. This is an indication that the reactor could withstand and remove more substrate at high OLR. However, a decline in SS-FFBR performance to 89.6% resulted from increasing the OLR up to 11.7 kg COD/m^3^d at the lowest HRT. The reactor performance in terms of COD removal efficiency was higher compared to results obtained by the research team of Yu et al. [7]. In another study, Najafpour, Zinatizadeh, and Lee [20] achieved a 93.7% COD removal efficiency at an HRT of 40 h using a three-stage RBC. Resmi and Gopalakrishna [21] achieved a COD removal efficiency of 82% at 10.2 kg COD/m^3^d. Borghei, Borghei, Sharbatmaleki, and Pourrezaie [22] reached a COD removal of 89.80% at OLR of 4.50 kgCOD/m^3^d in an up-flow aerobic immobilized biomass (UAIB) reactor treating the synthetic sugar-manufacturing wastewater. Furthermore, Hassani, Borghei, Samadyar, and Ghanbari [23] achieved a COD removal of 87.60 at OLR of 9.08 kg COD/m^3^d in an MBBR treating industrial wastewater. In one of our studies conducted using a sequencing batch flexible fibre biofilm reactor (SB-FFBR), an excellent COD and TSS removal efficiency were reported to be 97.5% and 99.3%, respectively, at low OLR of 0.4 kg COD/ m^3^d, COD_in_ of 945 mg/L and retention time of 2 days [12]. Furthermore, in another study, we could reach an optimum COD removal efficiency at HRT of 8 h and CODin of 3922 mg/L in a single stage flexible fibre biofilm reactor (SS-FFBR) [8]. 

Furthermore, the effect of the OLR in terms of COD removal rate on the SS-FFBR system performance is presented in Figure 5b. From the figure, the COD removal rate increased linearly with increasing OLR at all HRTs studied. From the statistical analysis, the increase in OLR corresponding to the increase in CODin and reduction in HRT led to the increase in the COD removal rate. However, the upward trend was observed for the response at any HRT. This means the productivity of the system in terms of the amount of COD removed from wastewater at maximum OLR was high. The COD removal rate was increased from 1.08 to 10.67 kg COD/m^3^d when the OLR increased from 1.12 to 11.2 kg COD/m^3^d. The Figure clearly showed that even when the organic loading rate exceeded 11.7 kg COD/m^3^d, a high COD removal rate was observed, indicating the high system capacity. A similar trend also was reported in the literature [7,16,20]. The enhancement in biomass concentration obtained due to the presence of the flexible fibre was recognized as the main reason of better reactor performance at high organic loading rate. 

The influence of OLR on total suspended solids (TSS) content in the reactor is shown in Figure 5c. The results clearly revealed a very good performance of the SS-FFBR with respect to TSS removal efficiency. As shown in this Figure, with the increase in OLR, the TSS removal efficiency decreased. The highest TSS removal efficiency was achieved by reaching up to 96.7% at a low OLR corresponding to 1.145 kg COD/m^3^d and HRT of 16 h, whereas the TSS removal efficiency slightly decreased to around 93.6% when the OLR increased to 5.62 kg COD/m^3^ d at 16 h HRT. Such a good performance might be attributed to the low amount of total suspended solids in the influent wastewater. Generally, the TSS removal efficiency did not drop below 92% at both 12 and 16 h HRTs, even with the increase in OLR to 8 kg COD/m^3^d. Approximately, 89.7% TSS removal efficiency could be achieved at a OLR of 11.67 kg COD/m^3^d, which indicates that the SS-FFBR was highly effective for the treatment of raw milk processing wastewater at a short HRT. This might be due to high biomass sloughing and the high amount of food available in the reactor as the OLR increased. Raj and Murthy [24] observed a similar trend using trickling filter treating synthetic dairy wastewater. Considering the better performance of the SS-FFBR at high OLRs, this system can be categorized in the family of high rate systems.

Figure 5d illustrates the effect of the OLR on the TSS removal rate of the SS-FFBR system at different HRTs. It can be noted that at all HRTs, the TSS removal rate increased linearly with increasing OLR up to 11.67 kg COD/m^3^d. From this Figure, the TSS removal rate increased from 0.243 kg TSS/m^3^d to 3.00 kg TSS/m^3^d as the OLR increased from 1.145 to 11.67 kg COD/m^3^d. The OLR had a more significant influence on the TSS removed from the system. This indicates that the SS-FFBR is a practical alternative for the treatment of wastewaters with a high OLR compared to other processes. The reduction in amount of solids might be due to oxidation of the organic compounds.

The effect of OLR on the effluent turbidity was studied and shown in Figure 5e. From this Figure, it is seen that the effluent turbidity increases with the increase in OLR. At an OLR of less than 2.5 kg COD/m^3^d, the SS-FFBR generated a very high effluent turbidity ranged from 2.73 to 5.85 NTU. However, the effluent turbidity was increased to 44.6 NTU as the OLR increased to its maximum value of 11.67 kg COD/m^3^d. The increase in effluent turbidity was attributed to the increase in suspended solids in the effluent, which may be due to biomass sloughing. In general, the effluent turbidity results were very much lower than those obtained by Najafpour et al. using a three-staged aerobic rotating biological contactor (RBC) to treat food canning wastewater where minimum turbidity was reported to be 46 NTU at 40 h HRT [20].

The effect of OLR on the level of dissolved oxygen concentration (DO) in the reactor is shown in Figure 5f. There is a direct relation between the dissolved oxygen concentrations and low OLRs. At lower OLRs of 1.145, 1.66, and 2.5 kg COD/m^3^d, the DO concentration in the reactor was 5.18, 5.82, and 6.45 mg O_2_/L, respectively. The DO level, which was in a realistic range, representing sufficient aeration, does not show that the system had a deficiency for aeration even at high OLR. The reduction in the DO concentration in the reactor was due to the increase in the biomass activities in the reactor that increased the uptake of DO as the substrate concentration increased.

### 3.5. Effect of Suspended Solids Loading Rate (SLR)

The effect of the SLR on performance of the SS-FFBR with respect to TSS removal efficiency is presented in Figure 6a. The response was measured three times, and average value was reported. The TSS removal efficiency was dependent on the SLR, and it decreased with the increase in the SLR and decrease in the HRT. The maximum TSS removal efficiency of 96.7% was obtained at a SLR of 0.25 kg TSS/m^3^d and at 16 h HRT. However, there was slightly decreased TSS removal efficiency at HRT of 8 h. The TSS removal efficiency was 91.4, 90.2, and 89.4% when the SLR increased from 0.54, 1.34, and 3.00 kg TSS/m^3^d, respectively. These findings were considerably better than those achieved in the study by Najafpour, Zinatizadeh, and Lee [20]. The authors achieved only an 85% TSS removal efficiency at a low SLR of 1 kg TSS/m^3^d and HRT of 48 h, whereas the TSS removal efficiency dropped to 46% at 1.2 kg TSS/m^3^d SLR and 40 h HRT. 

Figure 6b shows the influence of the solids loading rate on the TSS removal rate at different HRTs. The SLR has a similar effect on the process responses to the OLR. It can be noted from this Figure that the TSS removal rate increases with increases in the SLR, and there is a strong correlation between these two factors at various HRTs. The TSS removal rate has a linear relationship with SLR. TSS removal rates were1.15, 1.78, and 3 kg TSS/m^3^d at SLRs of 1.23, 2.24, and 3.32 kg TSS/m^3^d and HRTs of 16, 12, and 8 h, respectively. It can be deduced that reactor productivity in terms of TSS removal rate was high as well. 

Effluent turbidity as a function of SLR for the SS-FFBR at various HRTs is shown in Figure 6c. An increase in the SLR results in an increase in the effluent turbidity. The minimum effluent turbidity in the range of 2.73 to 6.01 NTU was achieved at a low range of SLR below 1 kg TSS/m^3^d at various HRTs. However, the maximum effluent turbidity was 44.6 and 41.7 NTU at SLRs of 3.32 and 2.24 kg TSS/m^3^d HRTs of 8 and 12 h, respectively. Such sudden increase in the effluent turbidity was presumably due to the increase in the SLR and therefore the high amount of total suspended solid in the influent wastewater stream. In general, the proposed system indicated better performance as effluent turbidity compared to the other works [20]. These findings illustrate that the SS-FFBR could be a suitable alternative to other biofilm reactors, with a high effluent quality produced at short HRT and high SLR and OLR.

## 4. Conclusions

In this research, a good performance of the SS-FFBR is obtained with a high COD removal efficiency of 95% at a low influent COD concentration of 809 mg/L. The COD removal was slightly decreased to 91.7% as the influent COD concentration increased to nearly 4000 mg/L. The effect of OLR on the SS-FFBR performance was experimentally studied as parameters of TSS removal efficiency, DO, and turbidity. The SS-FFBR could increasingly tolerate high OLR, but with a corresponding slight decrease on the COD removal efficiency. This was observed even at the highest OLR of 11.7 kg COD/m^3^d, where the SS-FFBR achieved a good performance with 89.9% COD removal efficiency. From the findings, a linear relationship was observed between the OLR and COD removal rate. The COD removal rate was increased from 1.08 to 10.68 kg COD/m^3^d as the OLR increased from 1.145 to 11.7 kg COD/m^3^d. TSS removal efficiency of 96.7% was obtained at a low OLR of 1.145 kg COD/m^3^d. However, the removal efficiency of TSS declined to 89.7% at a maximum OLR of 11.67 kg COD/m^3^d. As a final conclusion, the SS-FFBR as a high rate system can be advisable for the treatment of various types of high strength industrial wastewaters.

## Figures and Tables

**Figure 1 membranes-13-00037-f001:**
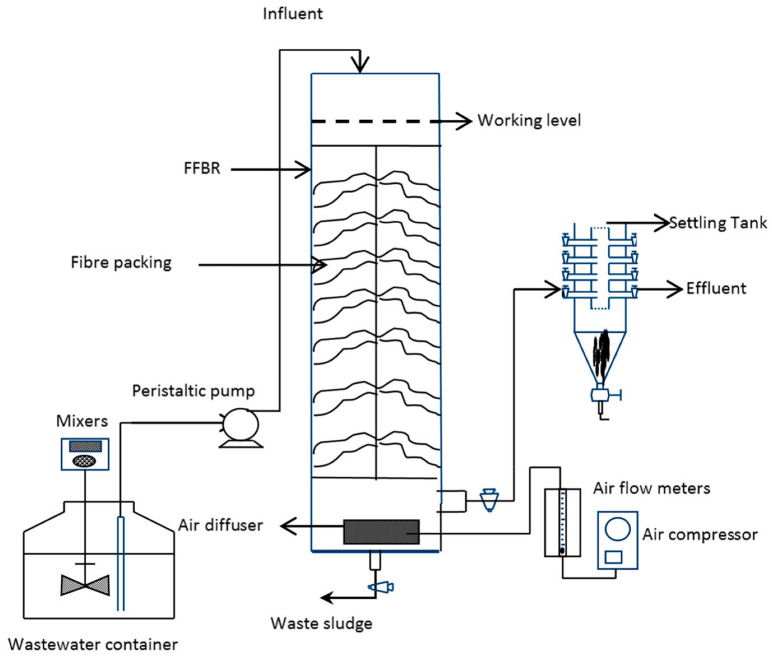
Schematic of SS-FFBR set-up used in this study.

**Figure 2 membranes-13-00037-f002:**
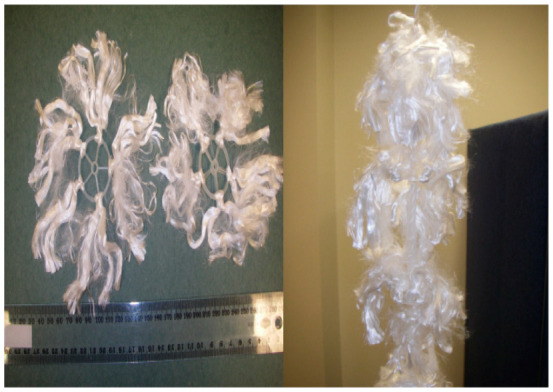
Photo of flexible fibre bundles.

**Figure 3 membranes-13-00037-f003:**
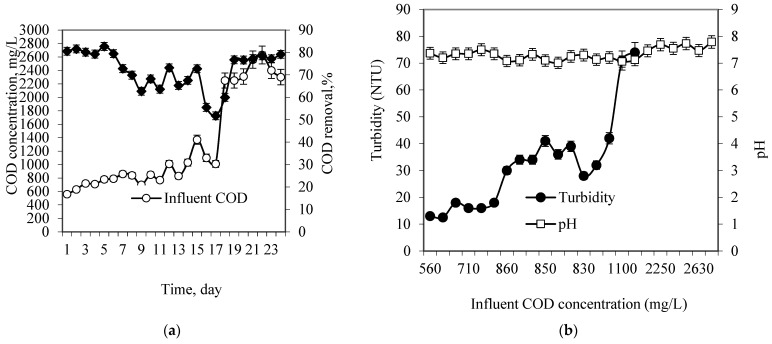
(**a**) COD removal efficiency; (**b**) effluent pH and turbidity variations; (**c**) influent and effluent VSS variations; (**d**) influent COD and DO levels during start-up period.

**Figure 4 membranes-13-00037-f004:**
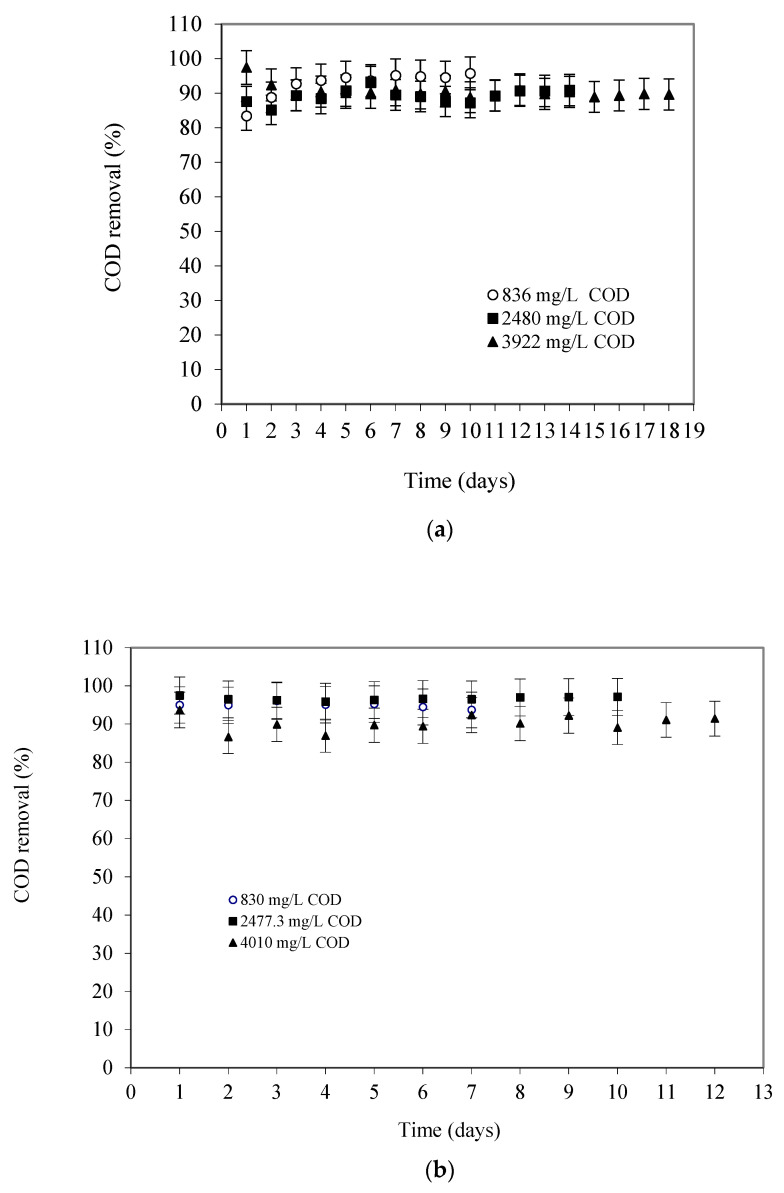
COD removal efficiency of SS-FFBR at various HRTs of (**a**) 8 h; (**b**) 12; (**c**) 16.

**Figure 5 membranes-13-00037-f005:**
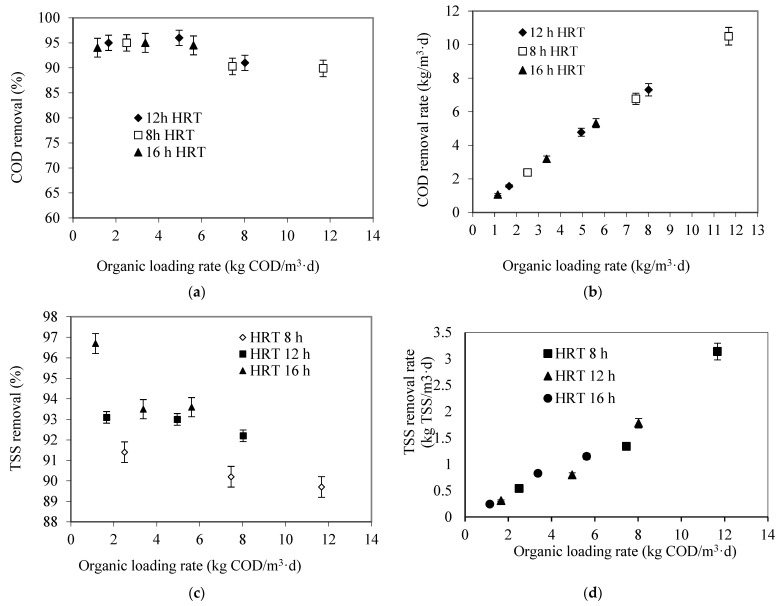
Effect of organic loading rate on (**a**) COD removal efficiency, (**b**) COD removal rate, (**c**) TSS removal efficiency, (**d**) TSS removal rate, (**e**) effluent turbidity, (**f**) DO concentration at different HRTs.

**Figure 6 membranes-13-00037-f006:**
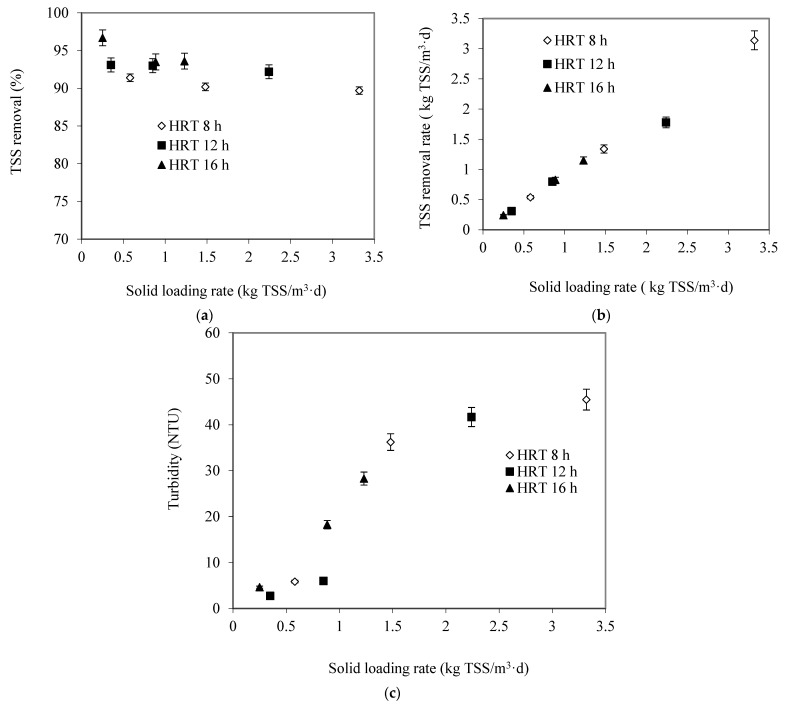
Effect of SLR on (**a**) TSS removal efficiency, (**b**) TSS removal rate, (**c**) effluent turbidity at different HRTs.

**Table 1 membranes-13-00037-t001:** Wastewater characteristics used in this study.

Parameters	Amount (mg/L)
COD	4000–14,250
BOD	3000–8910
pH	11.70
Total solids	5790–6380
Total suspended solid	1420–3540
Volatile suspended solid	1350–3480
Total nitrogen (as N)	N/A
Total Phosphate (as P)	37
Oil and grease	N/A

N/A = not nnalyzed.

**Table 2 membranes-13-00037-t002:** Specifications of the flexible fibre used as packing media in the SS-FFBR.

Parameters	Value
Bundle no.	7
Density, kg/m^3^	3
Specific gravity	1.02
Specific surface area, m^2^/m^3^	2164
Length of fibre, mm	75
Diameter of fibre, mm	0.07
Void ratio	0.99

**Table 3 membranes-13-00037-t003:** A list of operating and process variables used in this study.

Variables	Value
HRT, h	8–16
Influent flow rate, mL/min	16.7–8.3
COD_in,_ mg/L	800–4000
OLR, kg COD/m^3^d	2.4–12
SLR, kg/m^3^d	0.35–3.32

**Table 4 membranes-13-00037-t004:** SS-FFBR effluent quality under various HRTs.

Variables				Effluent Parameters			
Influent COD mg/L	HRT h	OLR kg COD/m^3^d	pH	DO mg/L	COD mg/L	TSS mg/L	Turbidity NTU
836	8	2.5	7.58	6	41	15	6.46
830	12	1.66	7.5	5.8	41.7	11.4	2.73
763	16	1.145	7.65	5.1	47.7	5	4.64
2480	8	7.44	7.73	3.45	238.5	40	36.2
2477	12	4.95	7.8	3	83.2	20	6
2266	16	3.37	7.9	3.19	111.5	25	18.25
3922	8	11.7	7.8	3.4	394.7	61.2	45.5
4010	12	8.02	7.5	3.6	352.8	51.4	41.7
3750	16	5.62	8.01	2.28	211.6	23.3	28.3

## Data Availability

The data presented in this study are available on request from the corresponding author.

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
