# Peer review of "Effect of Different Operational Conditions on the Treatment Performance of Milk Processing Wastewater (MPW) Using a Single Stage Flexible Fibre Biofilm Reactor (SS-FFBR)"

_membranes, 2022, doi:10.3390/membranes13010037_

Round 1

Reviewer 1 Report

The authors presented a study using flexible fibre biofilm reactor for the industrial wastewater treatment. This study benefits the readers who are interested in the biofilm reactor operations. However, some major revisions must be made before publishing.

First, I encourage the authors revise the Introduction and emphasize the difference and novelty of this study beyond the previous studies published from the same research group.

Below are the papers not cited in this paper.

Abdulgader, Mohamed, et al. "Application of response surface methodology (RSM) for process analysis and optimization of milk processing wastewater treatment using multistage flexible fiber biofilm reactor." Journal of Environmental Chemical Engineering 8.3 (2020): 103797.

Abdulgader, Mohamed, et al. "Treatment capacity of a novel flexible fibre biofilm bioreactor treating high-strength milk processing wastewater." Environmental Technology (2021): 1-17.

Abdulgader, Mohamed, et al. "Performance and kinetics analysis of an aerobic sequencing batch flexible fibre biofilm reactor for milk processing wastewater treatment." Journal of environmental management 255 (2020): 109793.

Please add these papers in your references and add the discussions of the research progress the authors achieved in this study. To my understanding, various operational conditions including HRT and COD loading have been discussed thoroughly in the previous papers using a similar wastewater and similar reactor set up.

Another concern is about the long HRT and corresponding suspended growth. In my opinion, the HRT was a bit longer in this study for a regular biofilm reactor operation. Therefore, suspended growth may outcompete the attached status for bacteria growth. It will affect the biofilm performance, and easily cause biomass loss when the environment changes. The authors also mentioned that in Line 235-236, Line 254-256, Line 269-271. I’d like to see the authors’ comments in the manuscript. Please also justify your designs for both biofilm and suspended growth.

Instead of a traditional biofilm reactor, this FFBR system is more likely similar to an IFAS. A significant amount of suspended biomass was existed in the reactor, unlike a traditional biofilm reactor. I’d like to have that stressed throughout the manuscript.

Based on above, to maintain the solids inventory and well mixing, the sludge should be recycled to achieve an ideal performance. the authors should comment it in the manuscript.

Some minor comments are below:

Section 2.2 – Please list the influent flow rate and also provide a list of operational conditions in this section including HRT, COD, etc (expand Line 148-149 into a list).

Line 144: Please use hydrogen chloride and sodium hydroxide instead of HCL and NaOH.

Line 150: Where did the DO measured? The biofilm carriers seem fluffy and will block the oxygen transfer to upper levels once the biofilm is established on the lower level. Can the authors comment on it?

Figure 1: Some of the texts on the figure are partially hidden. Please revise it.

Did the authors take samples of COD, and solids from biofilm reactor or the supernatant of the settling tank? Please clarify it.

How often was the sludge wasted from the bottom of the biofilm reactor? I also believe the WAS should be wasted in settling tank.

Please add units to all data and revise them throughout the manuscript. For example, in Line 231, 836 mg/L, 2480 mg/L, 3922 mg/L.

Line 166-167: As mentioned before, the coexisting of suspended biomass and biofilm will cause this problem. Besides, the addition of recycling stream will increase the COD removal. A shorter HRT is required for the growth of robust biofilms.

Results: The authors should provide a figure showing the COD and SS as the function of time from Day 1, different applied HRT should be marked in the same figure as well.

Figure 3a & 3c: Which day is the day 1? The legends of all secondary y axes disappear. Please correct it, as well as the same problem in other figures.

Line 341-342: A statistical analysis should be conducted to support this conclusion.

For the lines that connecting data points in Figure 5, 6, and even 4, they seem to be incorrect and meaningless. The curvature of these line generated automatically by the visualization tools such as excel provides incorrect meanings of the data. It should be corrected by a strict statistically analysis (or simply deleting them).

Author Response

Dear Editor, 

 I attached the rebuttal letter where I addressed all responses for both reviewers 

Reviewer 2 Report

The manuscript of the authors is devoted to the topical issue of the processing of waste products from the production of dairy plants and, without a doubt, has a high practical significance. It can also be noted that the manuscript is well suited to the subject of the journal. The positive side of the work is the use by the authors of real rather than model waste. On the other hand, I have a feeling that the authors in the manuscript presented a good technical report, but in my opinion the work lacks fundamentality. I would suggest adding characteristics of the membranes used in the work, biofilm fouling, electron micrographs, or something similar to dilute the same type of description of the technical features of the installation. For this reason, it is quite difficult to determine the general impression of the material.

General comments on the work are as follows:

In general, I would like to note the negligence in the design of the manuscript (different line spacing, etc.)

Authors should expand the introduction by citing more papers. the total number of references (11) shows that there are not enough of them.

Error bars should be shown in the figures.

In general, I believe that this work can be completed and it will be more interesting for a wider readership.

Author Response

All comments for both reviewers are addressed in one file

Round 2

Reviewer 1 Report

n/a

Reviewer 2 Report

this time the manuscript looks much better, I think it can be accepted in this form.